# Not all quantifiers are equal: Probing transformer-based language models' understanding of generalised quantifiers

**Tharindu Madusanka** and **Iqra Zahid** and **Hao Li**
**Ian Pratt-Hartmann** and **Riza Batista-Navarro**
Department of Computer Science
The University of Manchester

## Abstract

How do different generalised quantifiers affect the behaviour of transformer-based language models (TLMs)? The recent popularity of TLMs and the central role generalised quantifiers have traditionally played in linguistics and logic bring this question into particular focus. The current research investigating this subject has not utilised a task defined purely in a logical sense, and thus, has not captured the underlying logical significance of generalised quantifiers. Consequently, they have not answered the aforementioned question faithfully or adequately. Therefore, we investigate how different generalised quantifiers affect TLMs by employing a textual entailment problem defined in a purely logical sense, namely, model-checking with natural language. Our approach permits the automatic construction of datasets with respect to which we can assess the ability of TLMs to learn the meanings of generalised quantifiers. Our investigation reveals that TLMs generally can comprehend the logical semantics of the most common generalised quantifiers, but that distinct quantifiers influence TLMs in varying ways.

## 1 Introduction

Generalised quantifiers have been a topic of much interest for more than a century in logic and linguistics (Frege, 1882; Westerståhl, 1987; Gabbay et al., 1989; Mostowski, 1957). By capturing the interplay between quantity and cardinality, they provide a useful lens through which to understand human language and cognition (Troiani et al., 2009; Szymanik and Zajenkowski, 2010a). Since transformer-based language models (TLMs) strive to stimulate human-like language understanding (Vaswani et al., 2017; Devlin et al., 2018; Raffel et al., 2019; Ouyang et al., 2022; Chowdhery et al., 2022), it is essential to determine the extent to which they can comprehend generalised quantifiers. Assessing the depth of understanding that TLMs possess for any given concept is best achieved by evaluating their proficiency in applying it. In the case of generalised quantifiers, the most suitable evaluation task is textual entailment. This is particularly relevant because altering quantifiers can fundamentally change the logical inferences derived from a given text, reinforcing the integral role that quantifiers occupy within the scope of the textual entailment task.

When discussing entailment, it is vital to acknowledge two distinct strands of research in the literature. The first strand incorporates background knowledge and common sense into entailment, imbuing it with a probabilistic character (Bowman et al., 2015; Williams et al., 2018; Wang et al., 2019). The second strand examines textual entailment in a purely logical sense, eliminating the influence of background knowledge and common sense (Richardson and Sabharwal, 2021; Schlegel et al., 2022; Madusanka et al., 2023). While the first form of entailment proves beneficial for a multitude of practical applications, it is not ideal in an investigation centred on the impact of linguistic properties with logical significance, such as generalised quantifiers and negation. The empirical evaluation of linguistic constructs under this kind of entailment gets compromised due to its intricate association with other concepts. Consequently, it is challenging to differentiate the performance variation due to linguistic properties from those attributable to concepts like common sense and background knowledge. However, prior literature has only investigated generalised quantifiers in the context of entailment that incorporates background knowledge and common sense (Cui et al., 2022; Apidianaki and Garí Soler, 2021) and naturally suffers from the same predicament. The second strand of textual entailment by defining entailment in a purely logical sense circumvents the aforementioned shortcoming. Consequently, it offers a conducive environment for conducting evaluations

| **Structure in Natural Language:** *Talia, Hailee, Ava, Aria, Tony, Roger, Peter and Solomon are members of some group. Talia, Ava, and Solomon are bee-keepers. Hailee, Ava and Tony are scientists. Tony, Roger, Solomon, Ava, Aria and Hailee are Musicians. Roger, Solomon, Ava and Aria are guitarists. There are no designers in the group. Peter, Tony and Roger are artists. Only Solomon is an engineer.* | |
|---|---|
| **Sentence:** At least 3 musicians are guitarists **validity:** $True$ | **Sentence:** All bee-keepers are scientists **validity:** $False$ |

Figure 1: An instance of the model-checking with natural language problem, the sentence "At least 3 musicians are guitarists" is $True$ according to the structure since the set musicians $X = \{Roger, Solomon, Ava, Aria\}$ are also guitarists and $|X| \geq 3$. However, the sentence " All bee-keepers are scientists" are $False$ as the set of bee-keepers $\{Talia, Solomon\}$ are not scientists

centred around linguistic constructs.

The logical problem that is most suited to study the influence of language constructs of logical significance is that of model-checking: given a formula $\phi$ and a structure $\mathfrak{A}$, determine whether $\phi$ is true in $\mathfrak{A}$ ($\mathfrak{A} \models \phi$). In the context of natural language, we are interested in a variant of the model-checking problem where the structure and the formula are translated into natural language. An instance of model-checking with natural language problem is depicted in Figure 1. From a complexity-theoretic point-of-view, model-checking in most formal languages is, comparatively speaking, straightforward. Indeed the model-checking problem with a fixed number of free variables and a finite structure is in PTIME. This is in contrast to other logical problems, such as satisfiability, whose problems for various fragments of logic can belong to different computational complexity classes (Pratt-Hartmann, 2004; Pratt-Hartmann and Third, 2006). Yet, solving instances of the model-checking problem with natural language requires a comprehensive understanding of the logical semantics of the expressions involved. Thus, it provides an ideal test environment to faithfully evaluate the extent to which generalised quantifiers affect transformer-based language models.

In this study, we embark on an in-depth investigation into TLMs' understanding of generalised quantifiers utilising the model checking problem, juxtaposing this with cognitive science research on quantifier verification tasks (Szymanik and Zajenkowski, 2010a,b; McMillan et al., 2005). A critical part of our exploration involves the evaluation of pre-trained models prior to any fine-tuning. Thus, allowing us to discern whether any differences identified are intrinsic to the models themselves or introduced through the process of fine-tuning. Additionally, we consider the complexities arising from the integration of Boolean con-

junctions and negation with generalised quantifiers. This aspect of our study sheds light on the intricate dynamics that exist between these linguistic elements and the challenges they pose to TLMs. This comprehensive analysis paves the way for a more nuanced understanding of how TLMs handle intricate linguistic constructs such as generalised quantifiers.

The key contributions of the present research can be summarised as follows: (1) To the best of our knowledge, this study represents the first exploration into the effects of generalised quantifiers within a logical entailment context; (2) We analyse the effect on TLMs when quantifiers are paired with diverse logical constructs like negation and Boolean conjunctions; (3) We compare and contrast the behaviour of TLMs with quantifiers with that of quantifier verification experiments done with human beings; and (4) We delve into how well TLMs comprehend generalised quantifiers in a zero-shot context employing prompt engineering approaches such as chain-of-thought-prompting (Wei et al., 2022b) and provide comparisons between pre-trained and fine-tuned models.

## 2   Related Work

Our work follows the literature on probing how different linguistic properties affect the behaviour of neural approaches such as transformer-based language models (Madusanka et al., 2023; Clark et al., 2021; Buijtelaar and Pezzelle, 2023; Jawahar et al., 2019; Ettinger, 2020). Specifically, our investigation is closely related to the literature whose linguistic properties of interest are generalised quantifiers (Cui et al., 2022; Apidianaki and Garí Soler, 2021). Our exploration differentiates from prior research in two key ways. First, we explore generalised quantifiers employing a task that is defined purely in a logical sense. Thus, we provide a more faithful investigation into how TLMs comprehend

| GQ | Logical Denotation |
|---|---|
| All | $\{(X, Y) \mid X \subseteq Y \subseteq \mathfrak{A}\}$ |
| Some | $\{(X, Y) \mid X \cap Y \neq \emptyset \text{ and } X, Y \subseteq \mathfrak{A}\}$ |
| At least K | $\{(X, Y) \mid |X \cap Y| \geq K \text{ and } X, Y \subseteq \mathfrak{A}\}$ |
| At most K | $\{(X, Y) \mid |X \cap Y| \leq K \text{ and } X, Y \subseteq \mathfrak{A}\}$ |
| Less than K | $\{(X, Y) \mid |X \cap Y| < K \text{ and } X, Y \subseteq \mathfrak{A}\}$ |
| More than K | $\{(X, Y) \mid |X \cap Y| > K \text{ and } X, Y \subseteq \mathfrak{A}\}$ |
| K | $\{(X, Y) \mid |X \cap Y| = K \text{ and } X, Y \subseteq \mathfrak{A}\}$ |
| Most | $\{(X, Y) \mid |X \cap Y| > \frac{1}{2}|X - Y| \text{ and } X, Y \subseteq \mathfrak{A}\}$ |
| Few | $\{(X, Y) \mid |X \cap Y| < \frac{1}{2}|X - Y| \text{ and } X, Y \subseteq \mathfrak{A}\}$ |

Table 1: The generalised quantifiers (GQ) we used in our experimental setup, along with their logical denotation defined on some structure $\mathfrak{A}$.

generalised quantifiers. Second, our research also integrates a comprehensive analysis of how the interaction of negations and Boolean conjunctions with quantifiers influences TLMs' performance in a simple entailment task. We follow the logical denotations introduced in logical studies to formalised generalised quantifiers when formulating our task (Westerståhl, 1987; Mostowski, 1957; Gabbay et al., 1989; Fuhrken, 1970; Peters and Westerståhl, 2006) and draw parallels with cognitive science work on quantifier verification in our experimental setup (Szymanik and Zajenkowski, 2010b; McMillan et al., 2005; Szymanik et al., 2016).

Our evaluation scheme for evaluating TLMs in a zero-shot setting builds upon prior literature on prompt engineering (Brown et al., 2020; Kojima et al., 2022; Wei et al., 2022b). However, ours is the first literature evaluating TLMs on the model-checking problem in zero-shot settings.

## 3 Methodology

### 3.1 Language Fragments and Generalised Quantifiers

We define a language fragment to be a set of sentence forms equipped with semantics translating those sentences to some formal system such as first-order logic (Pratt-Hartmann, 2004) and perhaps the simplest way to define a language fragment is via a finite set of sentence templates. A sentence template is a sentence in which certain open-class words have been replaced by schematic variables. For example, "All $A$s are $B$s" is a sentence template where $A$ and $B$ substitute ordinary nouns (e.g., artist, musician beekeeper, ...), and by substituting $A$ and $B$ with such nouns, we can formulate sentences such as "All musicians are artists". Due to the formal structure that exists in language frag-

ments, a set of sentence templates is a natural way of representing them. For example the Aristotelian syllogism (Smith et al., 1989) can be defined using the following set of templates,

*All As are Bs*          *Some As are As*
*No A is a B*          *Some B are not Bs*

In this work of literature, we employ a slightly extended version of the Aristotelian syllogistic to allow negations at the subject, (e.g., Some non-musicians are beekeepers) and generalised quantifiers when generating sentences.

Generalised quantifiers define the semantics of sentences that include them in terms of relations between subsets of the structure (Szymanik et al., 2013). Consider for example "All musicians are artists". The determiner phrase "All" in this sentence specifies a relation between the set of musicians and the set of artists, namely that the former is a subset of the latter. More generally, "All" in a structure $\mathfrak{A}$ expresses the binary quantifier:

$$\{(X, Y) \mid X \subseteq Y \subseteq \mathfrak{A}\}$$

This idea can be generalised to accommodate other quantifiers. Consider the sentence "At least $K$ musicians are artists" where $K \in \mathbb{N}$. The phrase "At least $K$" likewise expresses a relation between the set of musicians and the set of artists, namely the cardinality of their intersection is at least $K$, that is, "At least $K$"in $\mathfrak{A}$ expresses the binary quantifier:

$$\{(X, Y) \mid |X \cap Y| \geq K \text{ and } X, Y \subseteq \mathfrak{A}\}$$

In our scholarly inquiry, we examine logical quantifiers such as "All", numerical quantifiers such as "At least $K$" and propositional quantifiers such as "Most". The quantifiers employed, and

their logical denotation on structure $\mathfrak{A}$ are depicted in Table 1. We utilise these generalised quantifiers when defining language fragments for sentence generation. Let $\mathcal{T}_Q$ be the sentence template which defines the language fragment corresponding to the quantifier $Q$. For example, consider the quantifier "All", the corresponding template $\mathcal{T}_{All}$ takes the form of "All (non-)$A$s are (not) $B$s" where $A$ and $B$ are replaced by ordinary nouns. Appendix A depicts the sentence templates used to define language fragments for each of the quantifiers.

## 3.2 Data Construction

---

**Algorithm 1** Data Construction - Model checking with Generalised Quantifiers

---

**Input :** The Quantifier $Q$ and corresponding sentence template $\mathcal{T}_Q$, a natural language template $\mathcal{M}$ to convert the structure to natural language, the vocabulary of proper nouns $\overline{D}$ and ordinary nouns $\overline{P}$, minimum and maximum number of domain elements $d^{min}$ and $d^{max}$, minimum and maximum number of predicates $p^{min}$ and $p^{max}$

**Output :** model checking dataset $\mathcal{D}$

1: $\mathcal{D} \leftarrow \{\}$
2: **repeat**
3:    $D, P \leftarrow$ sample from vocabularies $\overline{D}$ and $\overline{P}$ such that $d^{min} \leq |D| \leq d^{max}$, $p^{min} \leq |P| \leq p^{max}$
4:    $A, B \leftarrow$ sample two predicates from $P$
5:    $\overline{A}, \overline{B} \leftarrow$ negate $A, B$ with $p_{neg}$
6:    $s \leftarrow$ substitute predicates $\overline{A}$ and $\overline{B}$ for schematic variables in the template $\mathcal{T}_Q$
7:    $\ell \leftarrow$ sample from $\{True, False\}$
8:    **repeat**
9:       $\mathfrak{A} \leftarrow$ generate structure randomly using $(D, P)$
10:      $\bar{\ell} \leftarrow$ MODELCHECKER($\mathfrak{A}, (Q, \overline{A}, \overline{B})$)
11:    **until** $\ell = \bar{\ell}$ '
12:    $M \leftarrow$ translates $\mathfrak{A}$ to natural language using the template $\mathcal{M}$
13:    $\mathcal{D} \leftarrow \mathcal{D} \cup \{M, s, \ell\}$
14: **until** *stop condition is met*

---

We develop a data construction algorithm (Algorithm 1) to construct a balanced dataset free from easily exploitable trivial linguistic patterns. The algorithm constructs a set of triplets $(M, s, \ell)$, where $M$ is the natural language translation of the structure, $s$ is a sentence of the relevant fragment $\mathcal{T}_Q$ and $\ell$ is a label ($True/False$) specifying whether $s$ is

true in $M$. To construct $(M, s, \ell)$, apart from $\mathcal{T}_Q$, the algorithm takes the vocabularies $\overline{D}, \overline{P}$ also as inputs. The vocabulary $\overline{D}$ comprises proper nouns employed to characterise domain elements, while the vocabulary $\overline{P}$ comprises ordinary nouns that characterise predicates. We draw a random sample of elements $D$ and $P$, from vocabularies $\overline{D}$ and $\overline{P}$ to construct the structure. Two random nouns are sampled from $P$, each is then negated with probability $p_{neg}$, and these are finally substituted for the two schematic variables in the template $\mathcal{T}_Q$ to form the sentence $s$.

Given (probably negated) words $\overline{A}, \overline{B}$, a generalised quantifier $Q$ and a structure $\mathfrak{A}$, the model-checker determines $\mathfrak{A} \models s$, where $s$ is the sentence formed by substituting $\overline{A}, \overline{B}$ for schematic variables in the templates $\mathcal{T}_Q$. This involves first determining the extensions of $\overline{A}$ and $\overline{B}$ in $\mathfrak{A}$ and then applying the meaning of generalised quantifier $Q$ to these sets. Consider the example put forth in the section 1, $(Q, \overline{A}, \overline{B})$ corresponding to the sentence "All bee-keepers are scientists" is $(All, beekeepers, scientist)$. the model-checker determines the extensions of $beekeepers$ and $scientists$ in the corresponding structure $\mathfrak{A}$ to be $\{Talia, Ava, Solomon\}$ and $\{Hailee, Ava, Tony\}$, respectively. The quantifier $All$ dictates that in order for the sentence to be $True$, the former needs to be a subset of the latter. However, the set $\{Talia, Ava, Solomon\} \not\subseteq \{Hailee, Ava, Tony\}$, thus, the model-checker assigns $False$ as the validity label $\bar{\ell}$.

This setup with relative ease can be extended when Boolean conjunctions are introduced to the sentences. Consider, for instance, a sentence pair $s_1$ and $s_2$, formed using the predicates $(\overline{A}_1, \overline{B}_1)$ and $(\overline{A}_2, \overline{B}_2)$, respectively, for some quantifier $Q$, merged using Boolean conjunction $\odot \in \{\wedge, \vee\}$. To adapt to this scenario, the algorithm can be augmented by effecting a simple modification to step 10, transforming it into $\bar{\ell} \leftarrow$ MODELCHECKER($\mathfrak{A}, (Q, \overline{A}_1, \overline{B}_1)$) $\odot$ MODELCHECKER($\mathfrak{A}, (Q, \overline{A}_2, \overline{B}_2)$).

## 3.3 Prompts for Zero-shot Evaluation

Given that transformer-based language models undergo pre-training through a certain form of language modelling objective, the most common approach to evaluate these models in the zero-shot setting is by employing prompt engineering (Brown et al., 2020). Consequently, we formulate prompts

following a template-based strategy, utilising the constructed tuples $(M, s, \ell)$. We adopt two distinct types of templates. The first adheres to a more traditional form of prompting, which we refer to as standard prompting. The second type of template is based on the concept of chain-of-thought prompting (Wei et al., 2022b). Chain-of-thought prompting is a technique in which an example problem instance, accompanied by an explanation of the underlying thought process, is used to guide the model towards generating more precise responses. We depict the exact templates in Appendix A.

## 4 Experimental Setup

### 4.1 Transformer-based language models

To explore the transformer-based language models' ability to comprehend different generalised quantifiers, we employ a set of TLMs that have a proven track record in textual entailment problems, namely, T5, Flan-T5, DeBERTa, LLaMA and GPT.

**T5** Following the prior work on textual entailment defined purely in a logical sense (Richardson and Sabharwal, 2021; Tafjord et al., 2021; Madusanka et al., 2023), we utilise the T5 model in our experimental setup as one of the baseline models. The T5 model (Raffel et al., 2019) employs a unified text-to-text format where all inputs and outputs are textual strings. We fine-tune the T5-large model with 770M parameters to perform the model-checking task.

**Flan-T5** Fine-tuned Language Net (Chung et al., 2022), also known as Flan, is based on instruction fine-tuning (Wei et al., 2022a) with the objective of making the transformer model generalise better to unseen tasks. The Flan-T5 model, considered to be an improvement to T5, applies instruction fine-tuning on the T5-model family. Thus, we primarily centred our experimental setup around the Flan-T5 model. We fine-tune the Flan-T5-large model with 770M parameters and utilise Flan-T5-base with 220M parameters, Flan-T5-large, Flan-T5-xl with 3B parameters and Flan-T5-xxl with 11B parameters in the zero-shot setting.

**DeBERTa-v3** Due to the recent success of the DeBERTa-v3 model (He et al., 2021) in solving natural language inference tasks, we utilise it as a baseline model. The DeBERTa architecture improves upon the BERT and Roberta models using a disentangled attention mechanism and enhanced mask

decoder. DeBerta-v3 further improves the architecture by utilising an ELECTRA-style pre-training with Gradient Disentangled Embedding Sharing. We fine-tune the DeBERTa-v3-large model with around 304M parameters.

**ChatGPT** Due to the recent success of ChatGPT in solving many natural language tasks in a zero-shot setting (Bang et al., 2023), we employ it in a similar context. Similar to InstructGPT (Ouyang et al., 2022), ChatGPT is trained to follow human instructions but follows a slightly different data collection approach.

**LLaMA** Considering that Flan-T5 and ChatGPT are trained to follow instructions, we decided to use a TLM which has not been explicitly trained to follow instructions as one of our baselines. Thus, we employ LLaMA-30B model in zero-shot settings. The LLaMA is said to outperform GPT-3 in most baselines and achieve comparable performance with respect to state-of-the-art TLMs (Touvron et al., 2023).

### 4.2 Dataset and Evaluation

To fine-tune and evaluate TLMs, we construct train and test sets with 72K and 36K unique problem instances with 8K and 4K data points for each generalised quantifier[1]. We arbitrarily select $[d^{min}, d^{max}] = [8, 14]$ and $[p^{min}, p^{max}] = [5, 10]$ when constructing problem instances. We construct a balanced dataset, and thus, we use accuracy as the main metric but chose to depict the overall precision, recall and f1-score to provide a more detailed analysis. We deem this setup answers the question, *"How do different quantifiers affect the behaviour of TLMs?"*. As the problem instances contain negations, our experiment will also provide insight on the effect of negation when intertwined with quantifiers on TLMs' understanding of language. We construct separate train and test sets with 72K and 36K problem instances with sentences containing Boolean conjunctions to answer the question, *"How do Boolean conjunctions affect the behaviour of TLMs when coupled with different quantifiers?"*. By evaluating these fine-tuned models against problem instances with higher $K$ values in the numerical quantifiers than that of the train set, we ask the question *"Do TLMs learn to comprehend the logical semantics of generalised quantifiers?"*. To answer the questions,

---

[1]The dataset and code available at `https://github.com/iTharindu/generalised-quantifiers-model-checking`

*"How do pre-trained TLMs comprehend different quantifiers?"* and *"Do they have any biases when performing a simple entailment task?"*, we evaluate TLMs in a zero-shot setting. We found that the problem instances with $[d^{min}, d^{max}] = [8, 14]$ and $[p^{min}, p^{max}] = [5, 10]$ are challenging for TLMs in zero-shot settings. Consequently, the use of the same test set did not yield any meaningful insights. Thus, we formulate a much simple problem instance with $[d^{min}, d^{max}] = [3, 6]$ and $[p^{min}, p^{max}] = [2, 4]$. A more detailed description of the dataset and fine-tuning is provided in Appendix B.

## 5 Results and Discussion

**The ability of transformer-based language models to solve instances of the model-checking problem is differentially influenced by various generalised quantifiers**. As demonstrated in Table 2, TLMs appear to encounter the most difficulty with proportional quantifiers such as "Most" and "Few". Interestingly, this empirical observation aligns with cognitive science research, which also highlights the complexities faced by humans in interpreting proportional quantifiers (Szymanik and Zajenkowski, 2010a; McMillan et al., 2005; Troiani et al., 2009). In addition, the performance related to the quantifier "K" is notably lower in comparison to other numerical quantifiers. A sentence incorporating the quantifier "K" is probabilistically more likely to be $False$ given a random structure. Therefore, in a balanced dataset, the determination of the truth value of a sentence containing the quantifier "K" necessitates a more detailed examination compared to other numerical quantifiers considered in this study. However, as illustrated in Figure 2, given an adequate number of training steps, all TLMs attain satisfactory performance levels across all quantifiers. Moreover, the newer TLM models, such as DeBERTa-v3 and Flan-T5, exhibit a faster convergence rate compared to T5.

As expressed by their precision and recall values, TLMs often predict $True$ for quantifiers such as "All" and "Less than K", but often predict $False$ with respect to quantifiers like "Some" and "More than K". We attribute this to be an overcorrection introduced during fine-tuning. Consider the sentence with the quantifier "Less than K": "Less than K artists are engineers". This sentence is more likely to be $False$ in the context of the real world for $K$ values we consider in this study ($8 \leq K \leq 14$)

| GQ | ac | pr | re | f1 |
|---|---|---|---|---|
| All | 91.4 | 88.6 | 95.3 | 91.8 |
| Some | 92.2 | 96.8 | 88.0 | 92.2 |
| At least K | 94.2 | 97.7 | 90.4 | 93.9 |
| At most K | 96.6 | 96.5 | 96.6 | 96.5 |
| Less than K | 95.1 | 93.4 | 97.2 | 95.3 |
| More than K | 94.9 | 96.9 | 93.0 | 94.9 |
| K | 90.8 | 86.7 | 96.2 | 91.2 |
| Most | 90.4 | 92.9 | 87.3 | 90.0 |
| Few | 91.1 | 93.1 | 89.2 | 91.1 |

Table 2: The test scores for the `Flan-T5-large` model across various generalised quantifiers. The abbreviations *ac*, *pr*, *re* and *f1* denote accuracy, precision, recall and F1 score values.

since there are more than 14 artists who are engineers in the world. This proposition remains true even when negations are introduced to the sentences. Thus, we speculate that TLMs overcorrect during fine-tuning and predict $True$ or $False$ accordingly.

**TLMs show evidence of learning to understand the logical semantics of generalised quantifiers**. As illustrated in Figure 3, when tested with a dataset containing higher $K$ values than that of the train set, the accuracy of TLMs only decreases slightly for all numerical quantifiers. Therefore, we posit that TLMs possess the capacity to learn the logical semantics associated with generalised quantifiers. Our conclusions regarding generalisation bear resemblances to prior work conducted on model-checking with natural language (Madusanka et al., 2023). Their research also supports the premise that TLMs are capable of comprehending the logical semantics of natural language. Additionally, we highlight the contrast between the demonstrated ability of TLMs to generalise in the context of model-checking problems, and their apparent lack of such generalisation when solving satisfiability problems (Schlegel et al., 2022; Richardson and Sabharwal, 2021). We hypothesise that this distinction is due to the different complexity levels associated with these two types of problems and the necessity to understand complex inference rules when solving satisfiability problems.

**The Boolean conjunctions have a significant effect, while negation has much less effect on fine-tuned TLMs when coupled with generalised quantifiers.** As demonstrated in Table 3, it is apparent that fine-tuned TLMs possess the capacity to

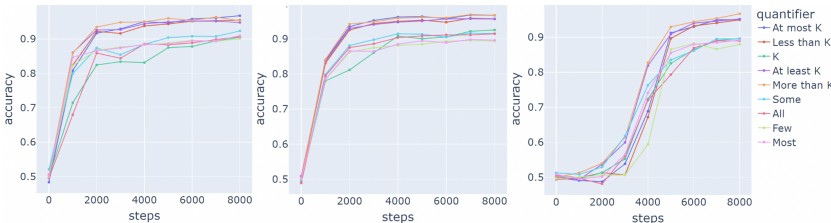

Figure 2: The rate of convergence of (a) `Flan-T5-large` and (b) `DeBERTa-v3-large` and (c) `T5-large` models break down based on different quantifiers

| | ac | | pr | | re | | f1 | |
|---|---|---|---|---|---|---|---|---|
| | $AND$ | $OR$ | $AND$ | $OR$ | $AND$ | OR | $AND$ | $OR$ |
| All | 91.0 | 83.7 | 89.0 | 80.9 | 93.8 | 89.5 | 91.3 | 84.9 |
| Some | 83.4 | 92.3 | 89.0 | 96.4 | 75.9 | 88.2 | 81.9 | 92.1 |
| At least K | 90.9 | 86.9 | 93.3 | 84.7 | 88.8 | 89.4 | 91.0 | 87.0 |
| At most K | 86.4 | 93.2 | 95.4 | 95.1 | 76.5 | 91.0 | 84.9 | 93.0 |
| K | 87.5 | 74.2 | 88.9 | 69.7 | 85.7 | 86.3 | 87.3 | 77.1 |
| Less than K | 88.5 | 90.6 | 92.3 | 90.3 | 84.0 | 91.0 | 87.9 | 90.7 |
| More than K | 92.5 | 88.5 | 93.2 | 85.4 | 91.6 | 92.9 | 92.4 | 89.0 |
| Most | 82.2 | 81.3 | 82.6 | 80.7 | 83.0 | 83.0 | 82.8 | 81.8 |
| Few | 82.2 | 81.0 | 83.4 | 81.8 | 80.1 | 79.0 | 81.7 | 80.4 |

Table 3: The test accuracy values for the `Flan-T5-large` model across various generalised quantifiers, broken down based on the Boolean conjunction ($AND$, $OR$) in the sentence. The abbreviations *ac*, *pr*, *re* and *f1* denote accuracy, precision, recall and F1 score values.

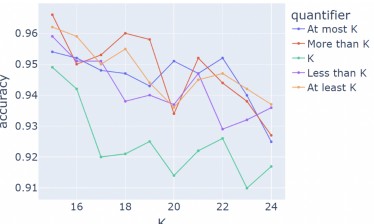

Figure 3: The accuracy value when tested against problem instances with sentences containing higher $K$ values than that of the train set. The results are broken down based on the numerical quantifier.

comprehend negations. In contrast, as depicted in Figure 4 (c), prior to the fine-tuning process, negations exert a considerable influence. Consequently, we propose that fine-tuning plays a significant role in enhancing the ability of TLMs to understand negations. The inclusion of Boolean conjunctions significantly reduces the accuracy for all quantifiers. Noticeably, quantifiers for whom TLMs tend to predict $True$ often tend to have higher accuracy in the context of the $OR$ operation compared to $AND$ and vice-versa. In our findings, we discovered that quantifiers for which TLMs frequently predict the label as $True$ also display elevated recall values for $OR$ operations and diminished recall for $AND$ operations. Conversely, quantifiers that TLMs often predict as $False$ exhibit higher precision for $AND$ operations and lower precision for $OR$ operations.

**The number of parameters, training process and type of prompting can influence the TLMs' performance when solving model-checking problem instances in zero-shot settings**. The performance for Flan-T5 models exhibits the power law relationship with the number of parameters, as illustrated in Figure 4 (a). This empirical finding is consistent with prior research analysing language models' performance variation with fac-

tors such as the number of parameters, dataset size and computational resources (Kaplan et al., 2020). However, upon breaking down the performance metrics based on the quantifiers, the resulting graph (Figure 4 (b)) is observed to be less uniform compared to the representation of overall performance. We attribute this behaviour to the inherent probabilistic aspect of the predictions formulated by TLMs since language models are trained to find the most probable next word given a set of words. This probabilistic nature of language models can lead to inaccurate predictions, especially in a logical context.

The Flan-T5 model with fewer parameters outperformed the ChatGPT and LLaMA models in a zero-shot setting, as depicted in Table 5. This phenomenon is unsurprising since Flan-based models are very effective in tasks naturally verbalised as instructions due to their employment of instruction fine-tuning (Wei et al., 2022a). Upon contrasting the efficiency of ChatGPT and Flan-T5 models in the context of standard and chain-of-thought prompting techniques, it is observed that the discrepancy in accuracy metrics across these two distinct prompting methodologies is not substantial. However, the LLaMA model generated both $True$ and $False$ when generating the label when standard prompting is used, failing to follow the instruction properly. We attribute this failure in

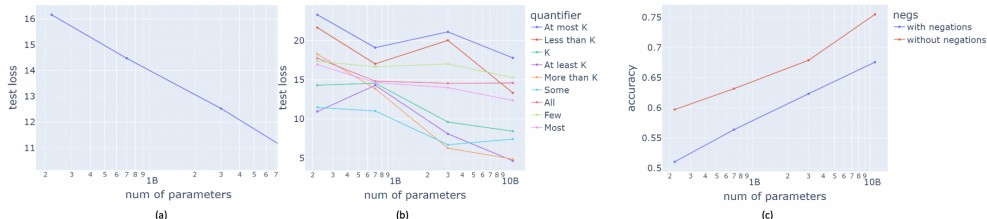

Figure 4: The (a) overall test loss and (b) test loss break down based on quantifier and (c) accuracy values breakdown based on the availability of negations for the Flan-T5 model with a different number of parameters in zero-shot settings, the number of parameters variates from $220M$ to $11B$.

| GQ | $s^0n^0$ | $s^0n^1$ | $s^1n^0$ | $s^1n^1$ |
|---|---|---|---|---|
| All | 91.6 | 94.7 | 89.0 | 89.6 |
| Some | 95.3 | 91.8 | 91.9 | 89.7 |
| At least K | 94.7 | 94.6 | 93.9 | 94.6 |
| At most K | 96.2 | 97.0 | 96.5 | 96.6 |
| Less than K | 96.1 | 95.5 | 95.7 | 95.0 |
| More than K | 95.6 | 95.2 | 95.4 | 94.9 |
| K | 90.9 | 91.8 | 91.5 | 89.7 |
| Most | 93.6 | 92.4 | 88.1 | 88.6 |
| Few | 91.7 | 92.5 | 87.4 | 89.3 |

Table 4: The test accuracy values for the `Flan-T5-large` model across various generalised quantifiers breakdown based on the negations in the sentence. The $s, n$ denote subject and predicate nominative, $1, 0$ denotes having or not having a negation at $s, n$. For example $s^0n^1$ denote no negation at subject and negation at predicate nominative.

|  | ChatGPT | | Flan-T5 | | LLaMA |
|---|---|---|---|---|---|
|  | $st$ | $ch$ | $st$ | $ch$ | $ch$ |
| All | 57.8 | 59.3 | 59.5 | 65.6 | 50.5 |
| Some | 77.7 | 80.0 | 79.4 | 79.1 | 58.0 |
| At least K | 84.1 | 80.0 | 87.1 | 87.1 | 67.3 |
| At most K | 42.9 | 42.2 | 50.6 | 46.9 | 45.3 |
| Less than K | 48.5 | 44.4 | 63.0 | 64.6 | 49.2 |
| More than K | 82.6 | 75.9 | 86.5 | 88.2 | 53.2 |
| K | 68.1 | 70.1 | 76.6 | 70.5 | 53.4 |
| Most | 54.6 | 65.4 | 65.7 | 69.1 | 57.9 |
| Few | 44.4 | 43.2 | 57.6 | 55.9 | 46.4 |
| Overall | 62.3 | 62.3 | 69.6 | 69.9 | 53.5 |

Table 5: The test accuracy values for the `ChatGPT`, `Flan-T5-xxl` and `LLaMA-30B` model in zero shot settings, $st$ denotes standard-prompting approach while $ch$ denotes the chain-of-thought prompting approach.

the LLaMA model to its training process, which, unlike the other two models, is not trained to follow instructions. When subjected to the chain-of-thought prompting approach, the LLaMA model displayed more consistency, generating either a $True$ or $False$ label. Thus, we infer that the inclusion of examples assists the LLaMA model in generating more concise outputs.

**Accuracy values for TLMs in zero-shot settings vary drastically with different quantifiers.** As depicted in Table 5, TLMs struggle with numerical quantifiers whose cardinality of intersection has an upper bound, such as "At most K" and "Less than K". This diminished performance can be attributed primarily to the TLMs' tendency to predict the label $False$ more frequently in sentences incorporating these specific quantifiers. We hypothesise this phenomenon is due to two factors. First, the background knowledge already embedded in TLMs from pretraining. As indicated previously, the sentences containing the above quantifiers coupled with a low $K$ value are often $False$ in a real-world

scenario. Second, the prior cognitive research on quantifiers has demonstrated that quantifiers with a downward monotone, such as "Few" or "Less than K", present more processing challenges for humans compared to those with an upward monotone, such as "Most" and "More than K" (Geurts and van der Slik, 2005; Zeijlstra, 2020; Agmon et al., 2019). Since TLMs are trained on human-generated data, it is highly likely that these models have incorporated this cognitive trait into their understanding of language, which, in turn, affects their responses. Moreover, a deeper analysis of the answers generated through chain-of-thought prompting revealed that even when the predicted label is correct, the overall answer is often incoherent. This coherence deficit in TLMs, coupled with their difficulties in handling certain quantifiers, suggests that these models are yet to achieve proficiency in learning even the simplest inferential rules.

## 6 Conclusion

We investigated how generalised quantifiers affect the behaviour of transformer-based language mod-

els by employing the problem of model-checking. We found that different generalised quantifiers affect TLMs in varying ways when solving model-checking problems in both fine-tuned and zero-shot settings. Based on empirical findings on generalisation, we posited TLMs can learn to understand the logical semantics of generalised quantifiers. Moreover, our experimental setup in the zero-shot setting demonstrated that a multitude of factors, such as the training process, size of the models and type of prompts, can affect the ability of TLMs to solve a simple entailment task. Thus, a compelling avenue for future research is to probe how varying factors affect transformer-based language models when solving a more complex entailment task, like determining satisfiability.

## 7 Limitations

Due to the empirical nature of this study, it suffers from an inductive dilemma on three fronts. One, in the front of transformers, the second related to generalised quantifiers and the third in relation to prompts we explored in zero-shot settings. We explored several transformer-based language models that are in line with prior literature and probe how different generalised quantifiers affect their behaviour. Nonetheless, due to the empirical nature of this investigation, it is plausible that some TLM architectures could deviate from the behavioural norms discussed in this paper when interacting with generalised quantifiers. A similar limitation applies to the range of generalised quantifiers examined, as the ones employed in our study do not represent the entire spectrum of generalised quantifiers. In zero-shot settings, this limitation further extends to the prompt templates we employed. We consider two types of prompt templates, but there is a multitude of alternative ways prompts can be formulated by using the $(M, s, \ell)$ triplets.

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

## A  Appendix: Templates

### A.1  Sentence Templates

When constructing sentences, as mentioned in the methodology section, we employ sentence templates. Let $Q$ be a generalised quantifier, and $\mathcal{T}_Q$ be the sentence template for the corresponding quantifier $Q$. Then $\mathcal{T}_Q$ take the general form,

$$Q \text{ (non-) } A\text{s are (not) } B\text{s}$$

The inclusion of "non/not" is determined by the availability of the negations and $A, B$ are ordinary nouns. Consider the quantifier "At most $K$" for some natural number $K$. Then the corresponding sentence template takes the form "At most $K$ (non-) $A$s are (not) $B$s". Table 6 depicts sentence templates corresponding to quantifiers considered in this study.

### A.2  Prompt templates for Zero-shot settings

We employ the tuples $(M, s, \ell)$ to delineate prompts for the language modelling objective, providing a framework for evaluating the effectiveness of TLMs in zero-shot settings. As mentioned in the methodology section, we explored two types of prompts. One, we informally called standard prompts and the other is based on chain-of-thought-prompting. The standard prompting is conceptualised by the following template,
Q: Given the following scenario, $M$. Is the sentence $s$ $True$ or $False$ according to the scenario?
A:
   The chain-of-thought prompting employs an example problem instance with an explanation of the thought process, thereby facilitating a more precise response from TLMs. If we let $(M_e, s_e, \ell_e)$ represent this example problem instance and $E$ elucidate the thought process, the chain of thought prompting can then be defined using the template,
Q: Given the following scenario, $M_e$. Is the sentence $s_e$ $True$ or $False$ according to the scenario?
A: $\ell_e$. $E$
Q: Given the following scenario, $M$. Is the sentence $s$ $True$ or $False$ according to the scenario?
A:

| GQ | Sentece Template |
|---|---|
| All | All (non-) $A$s are (not) $B$s |
| Some | Some (non-) $A$s are (not) $B$s |
| At least K | At least $K$ (non-) $A$s are (not) $B$s |
| At most K | At most $K$ (non-) $A$s are (not) $B$s |
| Less than K | Less than $K$ (non-) $A$s are (not) $B$s |
| More than K | More than $K$ (non-) $A$s are (not) $B$s |
| K | $K$ (non-) $A$s are (not) $B$s |
| Most | Most (non-) $A$s are (not) $B$s |
| Few | Few (non-) $A$s are (not) $B$s |

Table 6: The generalised quantifiers (GQ) we used in our experimental setup, along with their sentence templates

| GQ | maximum | minimum | mean |
|---|---|---|---|
| All | 161 | 47 | 90.1 |
| Some | 165 | 48 | 89.7 |
| At least K | 160 | 47 | 92.4 |
| At most K | 162 | 48 | 92.2 |
| Less than K | 163 | 49 | 92.4 |
| More than K | 168 | 49 | 92.1 |
| K | 164 | 46 | 89.9 |
| Most | 162 | 45 | 89.8 |
| Few | 158 | 47 | 89.6 |

Table 7: minimum, maximum and mean number of words (tokens) in problem instances ($M + s$) when seperated by SPACE

## B  Appendix: Dataset and Training Details

### B.1  Dataset details

We utilised nine quantifiers when constructing sentences. We construct train and test sets with 72K and 36K unique data points with 8K and 4K data points for each quantifier for fine-tuning and evaluating TLMs. As the vocabulary, We employed the Richardson and Sabharwal (2021) vocabulary of nouns, which contains a list of professions (e.g., "artist", "doctor"), that we extended by adding more professions. The dataset contains an equal number of $True$ and $False$ problem instances for each generalised quantifier. When constructing the dataset for fine-tuning TLMs, we select $[d^{min}, d^{max}] = [8, 14]$ and $[p^{min}, p^{max}] = [5, 10]$. For sentences with numerical quantifiers, we select a $K$ values randomly from the range $[1, |D|]$, where $|D|$ denotes the number of domain elements selected when formulating the problem instance. The minimum, maximum and mean number of tokens for problem instances of each quantifier is depicted in Table 7. To evaluate TLMs' behaviour with boolean conjunctions, we also constructed separate train and test sets with 72K and 36K data points. The dataset contains an equal number of problem instances for each conjunction, and quantifier pair. Moreover, since the intention was to compare the effects of generalised quantifiers on transformer-based language models, so we decided to use the simplest form of language templates, i.e. syllogistic.

We also emphasise the rationality behind the iterative approach we used in constructing the data. An alternative way of constructing problem instances is to derive the label $\ell$ using the model-checker instead of iteratively creating structures to match a pre-defined label and a sentence. However, this alternative approach can induce easily exploitable patterns. Consider the quantifier "K", "All" and "Some". For a random structure, quantifiers "K" and "All" are more likely $False$, while the quantifier "Some" is probabilistically $True$.

### B.2  Fine-tuning Details

Formally, we define the task as a binary classification problem where the objective of the transformer-based language model is to predict the label $\ell$ ($True$ or $False$) given the natural language interpretation of the structure $M$ and the sentence $s$ as the inputs. We select and fine-tune three TLMs, namely T5, Flan-T5, and DeBERTa-v3, all of which have previously demonstrated their efficiency and reliability in resolving textual entailment tasks. According to prior literature, the performance of TLMs mostly depends on the pre-trained data, and size of the models rather than the architectural choice (Raffel et al., 2019; Kaplan et al., 2020). Moreover, the accuracy values yielded for all TLMs are similar. Thus, we expect

a similar behaviour for other TLM architectures as well. Since the TLMs achieve satisfactory accuracy and since the central research interest is to analyse the behaviour of TLMs rather than identifying the best-performing TLM, we do not perform any hyperparameter tuning. Moreover, exploring several different TLMs and performing hyperparameter tuning leaves a higher carbon footprint (Strubell et al., 2019). Due to the nature of the research question, we consider such an exploration unnecessary.

**Loss function and optimizer** We fine-tune each TLM to predict the label $\ell$ given the $(M, s)$ by reducing the binary cross entropy loss over the target using the ADAM (Kingma and Ba, 2015) optimizer.

**Batch size** Utilising gradient checkpointing for memory-efficient fine-tuning, we set the batch size to 36.

**Number of epochs** We fine-tune each TLM for 4 epochs, resulting in 8000 steps.

**maximum token length** We set the maximum token length to 512 since the maximum problem length is much lower than that, thus, we do not truncate the inputs.

**learning rate** We set the learninig rate of $1 \times 10^{-5}$

We utilise Huggingface (Wolf et al., 2019) implementation when experimenting with the TLM models we consider in this study.