# OpenReview forum: "Not all quantifiers are equal: Probing Transformer-based language models' understanding of generalised quantifiers"
_EMNLP/2023/Conference — EMNLP 2023 Main_

### Official Review · Reviewer_bRRx · 2023-08-04

**Soundness:** 4

**Excitement:**

3: Ambivalent: It has merits (e.g., it reports state-of-the-art results, the idea is nice), but there are key weaknesses (e.g., it describes incremental work), and it can significantly benefit from another round of revision. However, I won't object to accepting it if my co-reviewers champion it.

**Missing References:**

While I do value the use of primary source literature, there are two excellent surveys that could be cited:
- For the logical semantics of quantifiers, Peters and Westerstahl 2006 _Quantifiers in Language and Logic_.
- For the cognitive aspects, Szymanik 2016 _Quantifiers and Cognition_

Some possible inspirations for analyzing differences between quantifiers:
- For some properties of quantifiers (monotonicity, etc) and other measures of complexity that may be relevant: van de Pol et al 2023 and some references therein: https://www.sciencedirect.com/science/article/pii/S001002772200138X
- For the difficulty of negative monotone compared to upward monotone quantifiers: Agnon et al 2019 https://doi.org/10.5334/gjgl.770 , Guerts 2003 https://www.sciencedirect.com/science/article/pii/S0010027702001804 , Guerts 2005 https://academic.oup.com/jos/article-abstract/22/1/97/1686377?redirectedFrom=fulltext

**Paper Topic And Main Contributions:**

This paper studies the ability of language models to understand/learn the meanings of generalized quantifiers: the meanings of expressions like "All", "most", "at least n", and so on.  These expressions have been central in the study of natural language semantics for decades, since they have a precise but quite abstract semantics.  The paper uses both prompting in the zero-shot setting and fine-tuning to evaluate several different language models' abilities.

In general, they find some success, especially in the fine-tuning scenarios.  In the zero-shot scenarios, there are some differences between prompts ("standard" versus chain-of-thought) and also an effect of scale (among other factors).

These results are not uninteresting, and it is very good to probe language models for their understanding of so-called "logical semantics", since this is a form of meaning in natural language that is very precise and may be learnable from text.  There are some areas where I think the paper could benefit from a bit of re-framing.  And I also would have appreciated some analyses that focus on the differences between quantifiers more, as opposed to just differences between models.  In other words: are there facts about the quantifiers that also explain the differences?

REBUTTAL EDIT: I thank the authors for their rebuttal and have upgraded my reproducibility score in response, but am leaving my other scores unchanged.

**Questions For The Authors:**

- Which k (or multiple k) was used for "at least k" et al?  And was there an effect of accuracy on this k?
- Line 514-516: "We attribute this behaviour to the inherent probabilistic aspect of the predictions formulated by TLMs." Can you say more here?
- For fine-tuning: it wasn't clear to me whether you did "instruction-tuning", i.e. continued training as a language mode., but to predict the word True/False, or whether you trained a new classifier head for the labels.  Which one?
- Do you think factors like complexity or monotonicity may effect LM performance?  I'm particularly struck by "at most k" and "less than k" being harder than their "upward" counterparts; see references below for evidence that these are also harder for people.  Or are these ore likely to be artifacts in the model case?

**Reasons To Accept:**

- Uses logical semantics of natural language to test how well language models learn the meaning.  This is a very nice case study for these questions.
- Compares multiple prompting methods, and also does some fine-tuning experiments.
- The use of quantifiers enables careful control over the data-generation process (for model-checking), enabling balance among True/False examples, among other factors.
- Uses open-source LMs, in addition to API-only models.

**Reasons To Reject:**

- The analysis could be more thorough, in particular focusing on differences between different quantifiers.
- Some details about the experiments (values of k, exact nature of fine-tuning and prompting setup) are either missing or hard to find (in Appendices).
- Some of the framing is a bit hard to follow: most of the introduction talks about the task of inference, but then the actual task in the paper is verification ("model-checking").  I'm a fan of using the model-checking task, but it took awhile to figure out whether they were going to be using an inference format or not in the experiments, so I think some re-wording at the beginning would be welcome.

**Reproducibility:**

4: Could mostly reproduce the results, but there may be some variation because of sample variance or minor variations in their interpretation of the protocol or method.

**Reviewer Confidence:**

4: Quite sure. I tried to check the important points carefully. It's unlikely, though conceivable, that I missed something that should affect my ratings.

---

> ### Author Rebuttal · Authors · 2023-08-26
>
> Thank you for your comments.
>
> *“The analysis could be more thorough, in particular focusing on differences between different quantifiers.”*
>
> &nbsp;&nbsp;&nbsp;&nbsp;&nbsp;&nbsp; We have had to prioritise the discussion points in our analysis due to page constraints.
>
> *“Some details about the experiments (values of k, exact nature of fine-tuning and prompting setup) are either missing or hard to find (in Appendices).”*
>
> &nbsp;&nbsp;&nbsp;&nbsp;&nbsp;&nbsp; The value of K is expressed in line 861 in Appendix B.1. The K value is randomly sampled to be an integer between 1 and D where D is the number of domain elements in the problem instance. This is true for both fine-tuning and prompting setups. As explained in line 403 in the dataset and evaluation section, the problem instances for the prompting setup have fewer domain elements and predicates. Then, the final prompt is constructed using the template introduced in Appendix A.1.
>
> *“Some of the framing is a bit hard to follow: most of the introduction talks about the task of inference, but then the actual task in the paper is verification ("model-checking"). I'm a fan of using the model-checking task, but it took awhile to figure out whether they were going to be using an inference format or not in the experiments, so I think some re-wording at the beginning would be welcome.”*
>
> &nbsp;&nbsp;&nbsp;&nbsp;&nbsp;&nbsp; We attempted to make clear that the task is identifying whether a sentence is True or False according to some paragraph in natural language (logical structure in natural language). To perform the said problem the transformer model needs to comprehend logical semantics of natural language and perform very simplistic reasoning operations. Although this can, formally, presumably be characterised as an entailment problem, we accept that it would have been better to use the term “model-checking” more prominently. In the revised version, we will reword at the beginning to make it more clear.
>
> *“Which k (or multiple k) was used for "at least k" et al? And was there an effect of accuracy on this k?”*
>
> &nbsp;&nbsp;&nbsp;&nbsp;&nbsp;&nbsp; As mentioned above, the K value is selected randomly to be an integer between 1 and D where D is the number of domain elements in the problem instance. We did not explicitly test the effect of K values since the range of K values is not that high. Also, since a higher K value more often means higher domain elements in the problem instance, the problems with higher K values would be longer in length. Therefore, it would be difficult to decouple the effect of K values from the effect of the length of the text.
>
> *“Line 514-516: "We attribute this behaviour to the inherent probabilistic aspect of the predictions formulated by TLMs." Can you say more here?”*
>
> &nbsp;&nbsp;&nbsp;&nbsp;&nbsp;&nbsp; We meant that the language models are trained to find the most probable next word given a set of words. This probabilistic nature of language models can lead to inaccurate predictions, especially in a logical context. We would include the clarification in the revised version of the paper.
>
> *“For fine-tuning: it wasn't clear to me whether you did "instruction-tuning", i.e. continued training as a language model but to predict the word True/False, or whether you trained a new classifier head for the labels. Which one?”*
>
> &nbsp;&nbsp;&nbsp;&nbsp;&nbsp;&nbsp; We trained the language models to predict True or False with a sequence classification head; we did not perform instruction fine-tuning.
>
> *“Do you think factors like complexity or monotonicity may effect LM performance? I'm particularly struck by "at most k" and "less than k" being harder than their "upward" counterparts; see references below for evidence that these are also harder for people. Or are these ore likely to be artifacts in the model case?”*
>
> &nbsp;&nbsp;&nbsp;&nbsp;&nbsp;&nbsp; The difference in performance is evident in the zero-shot setting, and not so much in the fine-tuned version. Considering language models are trained on language corpora produced by humans, it is very likely that the pre-trained models inherit difficulties that humans possess when processing different quantifiers.
>
> Regarding comments on reproducibility: we are happy to share the code and data and intend to share it in the revised version.

---

### Official Review · Reviewer_SHqe · 2023-08-04

**Typos Grammar Style And Presentation Improvements:** No suggestions for improving the pres…
**Soundness:** 5

**Excitement:**

4: Strong: This paper deepens the understanding of some phenomenon or lowers the barriers to an existing research direction.

**Missing References:**

I did not notice any missing references

**Paper Topic And Main Contributions:**

The work aims to assess the ability of transformer-based language models (TLMs) to learn the meanings of generalised quantifiers using a logically-defined textual entailment problem (i.e., one that attempts to avoid the influence of background knowledge and common sense). The form of textual entailment used is model-checking of natural language statements, and the evaluation occurs prior to fine-tuning, so that any differences found are intrinsic to the models. Since the definitions of some generalized quantifiers involve negation and conjunction, the work contributes towards our knowledge of whether and how TLSs can deal with negation. The work is assessed by comparison with human understanding of quatifiers, and by using chain-of-thought prompting in a zero-shot setting.  The workd considers logical quantifiers, numerical quantifiers and propositional quantifiers.

**Questions For The Authors:**

No questions for the authors.

**Reasons To Accept:**

The paper considers and important, but little-researched topic, and demonstrates (i) a sound approach to the problems it considers; (ii) interesting results and (iii) convincing analyses -- especially the results shown in Tables 4 and 5 and their analyses.


**Reasons To Reject:**

I don't see any reason to reject the paper.


**Reproducibility:**

4: Could mostly reproduce the results, but there may be some variation because of sample variance or minor variations in their interpretation of the protocol or method.

**Reviewer Confidence:**

3: Pretty sure, but there's a chance I missed something. Although I have a good feel for this area in general, I did not carefully check the paper's details, e.g., the math, experimental design, or novelty.

---

> ### Author Rebuttal · Authors · 2023-08-26
>
> Thank you for your comments

---

### Official Review · Reviewer_sJL9 · 2023-08-05

**Soundness:** 3

**Excitement:**

3: Ambivalent: It has merits (e.g., it reports state-of-the-art results, the idea is nice), but there are key weaknesses (e.g., it describes incremental work), and it can significantly benefit from another round of revision. However, I won't object to accepting it if my co-reviewers champion it.

**Paper Topic And Main Contributions:**

This paper delves into the impact of different quantifiers on Transformer-based Language Models (TLMs) through the utilization of prompting strategies. The conducted experiments specifically address how the choice of models and experimental setup influences the performance in assessing TLMs' ability to comprehend generalized quantifiers. By exploring these factors, the study sheds light on the effectiveness of TLMs in dealing with linguistic challenges related to quantifiers.

**Reasons To Accept:**

The paper presents an extensive and well-written methodology section. The detailed explanation of the research approach allows for a clear understanding of the experimental setup. Furthermore, experiments with state-of-the-art models, such as T5, FlanT5, DeBERTa, ChatGPT, and LLaMA, provide comprehensive insights into how different transformer-based language models are affected by different quantifiers. Incorporating multiple models enhances the study's reliability and generalizability, allowing for robust comparisons and insightful conclusions.

**Reasons To Reject:**

This work fails to meet the Reproducibility Criteria of the EMNLP call:
- The absence of dataset and code availability hinders the ability of the research community to verify and reproduce the findings, undermining the credibility of the results.
- The lack of hyper-parameter tuning and arbitrary parameter selection introduces biases and unfairness in comparing models and settings, rendering the experimental setup unreliable.
- The need for dataset construction details, such as the ambiguous generation of noun vocabularies and missing label distribution information, raises concerns about data quality and potential biases in the evaluation.

The limited number of templates used for sentence generation hampers linguistic variability and could impact the results' validity and generalizability.

**Reproducibility:**

3: Could reproduce the results with some difficulty. The settings of parameters are underspecified or subjectively determined; the training/evaluation data are not widely available.

**Reviewer Confidence:**

3: Pretty sure, but there's a chance I missed something. Although I have a good feel for this area in general, I did not carefully check the paper's details, e.g., the math, experimental design, or novelty.

---

> ### Author Rebuttal · Authors · 2023-08-26
>
> Thank you for your comments.
>
> Regarding the comment on reproducibility: *“This work fails to meet the Reproducibility Criteria of the EMNLP call: The absence of dataset and code availability hinders the ability of the research community to verify and reproduce the findings, undermining the credibility of the results.”*
>
> &nbsp;&nbsp;&nbsp;&nbsp;&nbsp;&nbsp; We are more than happy to share the code and it was our intention to add the link to the code and data in the camera-ready version.
>
> &nbsp;&nbsp;&nbsp;&nbsp;&nbsp;&nbsp; Regarding data generation, it is worth noting that the specifications of the data sets, and indeed the algorithms to realise them---which are relatively straightforward---were given in full detail.
>
> *“The lack of hyper-parameter tuning and arbitrary parameter selection introduces biases and unfairness in comparing models and settings, rendering the experimental setup unreliable.”*
>
> &nbsp;&nbsp;&nbsp;&nbsp;&nbsp;&nbsp; We did not perform hyperparameter tuning of any of the transformer-based language models (TLMs) for the following reasons which were mentioned in Appendix B.2:
> 1. All TLMs when fine-tuned achieved adequate performance and did not exhibit much variation among themselves (variation in average accuracy between transformer models is less than 1.5%) in terms of their performance.
> 2. The primary point of this research is not to identify which specific TLM achieves the best performance but rather to identify how TLMs behave as a class of models. Thus, we believe that performing hyperparameter tuning is not necessary.
> 3. Moreover, exploring several different TLMs and performing hyperparameter tuning leaves a higher carbon footprint.
>
> &nbsp;&nbsp;&nbsp;&nbsp;&nbsp;&nbsp; Given the above, we decided to use the TLMs following their default parameter setup.
>
> Regarding the comment on data construction: *“The need for dataset construction details, such as the ambiguous generation of noun vocabularies and missing label distribution information, raises concerns about data quality and potential biases in the evaluation”*
>
> 1. Noun vocabulary -  We use the Richardson and Sabharwal (2021) vocabulary of nouns, which contains a list of professions (e.g., "artist", "doctor"), that we extended by adding more professions.
> 2. Label distribution - As mentioned in line 861 in Appendix B.1, the dataset contains an equal number of True and False problem instances for each quantifier.
>
> &nbsp;&nbsp;&nbsp;&nbsp;&nbsp;&nbsp; Other details such as the number of domain elements, the number of predicates, the K value range we explored, and the number of data points generated for each quantifier are mentioned in Appendix B.1.
>
> *"The limited number of templates used for sentence generation hampers linguistic variability and could impact the results' validity and generalizability."*
>
> &nbsp;&nbsp;&nbsp;&nbsp;&nbsp;&nbsp; The intention was to compare the effects of generalised quantifiers on transformer-based language models, so we decided to use the simplest form of language templates, i.e. syllogistic. This in line with prior work of research when analysing the effect of different language properties on transformers using synthetic data (Richardson et al. (2020), Richardson and Sabharwal (2021), Madusanka et al. (2023)). Moreover, the constructed sentences have a close resemblance to data used when performing quantifier verification on human beings.
>
> Reference
>
> 1. *Richardson, K. and Sabharwal, A., 2022, June. Pushing the limits of rule reasoning in transformers through natural language satisfiability. In Proceedings of the AAAI Conference on Artificial Intelligence (Vol. 36, No. 10, pp. 11209-11219).*
>
> 2. *Richardson, K., Hu, H., Moss, L. and Sabharwal, A., 2020, April. Probing natural language inference models through semantic fragments. In Proceedings of the AAAI Conference on Artificial Intelligence (Vol. 34, No. 05, pp. 8713-8721).*
>
> 3. *Madusanka, T., Batista-navarro, R. and Pratt-Hartmann, I., 2023, May. Identifying the limits of transformers when performing model-checking with natural language. In Proceedings of the 17th Conference of the European Chapter of the Association for Computational Linguistics (pp. 3521-3532).*

---

### Meta-Review · Area_Chair_wyBY · 2023-09-17

**Recommendation:** 5

**Metareview:**

The paper presents a probing study to explore the ability of LLMs to understand various types of generalized quantifiers. Reviewers and I agree about the methodological soundness of the work, in which state-of-the-art models, such as T5, FlanT5, DeBERTa, ChatGPT, and LLaMA, are tested. Moreover, the paper addresses a relatively little-studied topic in nowadays NLP and represents an interesting example of using formal semantics to explore the inferential competences of LLM.

---

### Decision · Program_Chairs · 2023-10-07

**Decision:**

Accept-Main

**Comment:**

The paper presents a probing study to explore the ability of LLMs to understand various types of generalized quantifiers. Reviewers and I agree about the methodological soundness of the work, in which state-of-the-art models, such as T5, FlanT5, DeBERTa, ChatGPT, and LLaMA, are tested. Moreover, the paper addresses a relatively little-studied topic in nowadays NLP and represents an interesting example of using formal semantics to explore the inferential competences of LLM.